# Comparative Analysis of Anticonvulsant Activity of *Trans* and *Cis* 5,5′-Diphenylhydantoin Schiff Bases

**DOI:** 10.3390/ijms242216071

**Published:** 2023-11-08

**Authors:** Jana Tchekalarova, Petar Todorov, Tsveta Stoyanova, Milena Atanasova

**Affiliations:** 1Institute of Neurobiology, Bulgarian Academy of Sciences, 1113 Sofia, Bulgaria; tzafti@abv.bg; 2Department of Organic Chemistry, University of Chemical Technology and Metallurgy, 1756 Sofia, Bulgaria; pepi_37@abv.bg; 3Department of Biology, Medical University of Pleven, 5800 Pleven, Bulgaria; milenaar2001@yahoo.com

**Keywords:** 5,5′-diphenylhydantoin Schiff bases, *trans–cis*, MES, KA SE, mice

## Abstract

Recently, the four 5,5′-diphenylhydantoin Schiff bases, possessing different aromatic species (**SB1-Ph, SB2-Ph, SB3-Ph and SB4-Ph**) were synthesized, characterized, and evaluated for anticonvulsant activity in combination with phenytoin. In the present study, the **SB1-Ph** and **SB4-Ph** compounds were selected, based on their anticonvulsant potency, and compared with their *cis* isomers, prepared after a one-hour exposure to the UV source, for their anticonvulsant potency in the maximal electroshock (MES) test and the kainate (KA)-induced status epilepticus (SE) test in mice. In the MES test, the *cis* **SB1-Ph** compound exhibited superior to phenytoin and *trans* isomer activity in the three tested doses, while the *cis* **SB4-Ph** compound entirely suppressed the electroshock-induced seizure spread at the highest dose of 40 mg/kg. Pretreatment with the *cis* **SB1-Ph** compound and the *cis* **SB4-Ph** at the doses of 40 mg/kg, respectively, for seven days, significantly attenuated the severity of KA SE compared to the matched control group pretreated with a vehicle, while phenytoin was ineffective in this test. The *cis* **SB4-Ph** but not the *cis* **SB1-Ph** demonstrated an antioxidant effect against the KA-induced SE in the hippocampus. Our results suggest that *trans–cis* conversion of 5,5′-diphenylhydantoin Schiff bases has potential against seizure spread in the MES test and mitigated the KA-induced SE. The antioxidant potency of *cis* **SB4-Ph** might be associated with its efficacy in mitigating the SE.

## 1. Introduction

Epilepsy is a chronic neurological disorder that affects approximately 1% of people worldwide [1]. Spontaneous recurrent seizures represent the main symptom of this disease. Pharmacotherapy with anticonvulsants is the method of choice for epileptic patients. Although there is extensive funding for developing antiepileptic drugs (ASMs), about 30% of patients are resistant to treatment [2]. The first generations of ASMs were characterized by severe side effects, whereas new-generation ASMs have an advantage with good tolerability and low capacity for drug interaction [3]. 

The first-line ASM phenytoin effectively blocks partial and tonic-clonic seizures [4]. However, this ASM has low solubility and side effects during treatments. Heterocyclic systems and their Schiff bases are an essential class of compounds that have piqued the interest of researchers due to their varied range of biological functions, including anti-inflammatory, anticonvulsant, analgesic, antimicrobial, anticancer, antioxidant, anthelmintic, and antidepressant activities [5,6,7]. Schiff bases, a universal pharmacophore, are studied in screening investigations [8]. These compounds have an imine or azomethine (-C=N-) functional group. They are condensation products of primary amines with carbonyl compounds, which are gaining importance in medicine and pharmacy due to their ease of synthesis and isolation. A heterocyclic system, such as phenytoin, combined with an azomethine functional group would have a synergistic effect and increase biological activity. Schiff bases tend to isomerize due to the imine group (-C=N-) giving two stereoisomers *cis* and *trans* (E and Z) isomers, and the formation of these stereoisomers can be controlled kinetically or thermodynamically. These compounds have the potential to be photosensitive, undergoing rapid reverse photoisomerization from the more stable *trans* isomer to the less stable *cis* isomer. They can employ this characteristic to control, functionalize, and alter numerous biological functions. As a result, as the Schiff bases’ molecular arrangement changes, the compounds’ bioactivity also changes [9].

Recently, the four 5,5′-diphenylhydantoin Schiff bases, containing aromatic species (**SB1-Ph, SB2-Ph, SB3-Ph and SB4-Ph**), were synthesized and their structure–property characterization was carried out by X-ray, optical, and electrochemical methods [10]. Furthermore, the four 3-amino-phenytoin Schiff base derivatives were explored alone and in combination with phenytoin against maximal electroshock (MES)-induced seizure spread in mice [11]. Taking into consideration the potential anticonvulsant properties of 3-amino-phenytoin Schiff base derivatives, in the present study, we aimed to compare the anticonvulsant activity of *trans* and *cis* 5,5′-diphenylhydantoin Schiff bases **SB1-Ph** and **SB4-Ph** against the MES-induced seizure spread. Furthermore, the potency to mitigate status epilepticus (SE) induced by kainic acid (KA) and oxidative stress in the hippocampus was evaluated after sub-chronic pretreatment with the more potent *cis* isomers of Schiff bases at the dose of 40 mg/kg, that was effective against tonic-clonic seizures in the MES test.

## 2. Results

### 2.1. Grip Strength and Rotarod

No significant effect on the neuromuscular tone, measured in the grip strength apparatus was detected for both the *trans*- and the *cis*-forms of the two Schiff bases (**SB1-Ph** and **SB4-Ph**), administered at doses of 10, 20 or 40 mg/kg [One-way ANOVA: F(2,23) = 0.543; *p* = 0.621—*trans*
**SB1-Ph**; F(2,23) = 0.893; *p* = 0.462—*cis* **SB1-Ph**; F(2,23) = 1.641; *p* = 0.092—*trans* **SB4-Ph** F(2,23) = 1.307; *p* = 0.312—*cis* **SB4-Ph**] (Table 1). In addition, the tested Schiff bases in *trans*- and *cis*-form, respectively, did not affect the motor coordination of mice, when tested in the three doses mentioned above, suggesting a lack of myorelaxant activity.

### 2.2. Maximal Electroshock Test

In the MES test, the *cis*-form of **SB1-Ph** but not the *trans* isomer showed protection against the MES-induced hindlimb tonic phase at the three tested doses (10, 20 and 40 mg/kg) (Fisher exact test: *p* = 0.003 compared to the control group) (Figure 1). This effect was comparable to that of the positive control phenytoin (*p* = 0.015 compared to control group). Similarly, the *cis*- but not the *trans*-form of **SB4-Ph** compound exhibited potency to suppress the MES-induced seizure spread (Fisher exact test: *p* = 0.003 **SB4-Ph**, 10 and 20 mg/kg compared to the control group). Notably, the *cis* isomer of **SB4-Ph** demonstrated 100% protection against tonic seizures at the highest dose of 40 mg/kg (*p* < 0.001, 40 mg/kg compared to the control group). No mortality of the *cis*-forms was observed except for **SB4-Ph** at a dose of 20 mg/kg with 16% mortality rate compared to the controls with 87% and 33% mortality rate for most of the *trans* isomers. 

### 2.3. Kainate-Induced Status Epilepticus

The mice from all groups were pretreated with the positive control phenytoin (Ph group, 20 mg/kg), *cis*-forms of **SB1-Ph** and **SB4-Ph**, respectively (40 mg/kg), i.p. for seven days to assess the efficacy of *cis* isomers of novel phenytoin-related Schiff bases to mitigate seizure intensity during the KA-induced SE as well as its consequences on oxidative stress in the hippocampus. The matched control group was pretreated with a vehicle for a week in the same manner before the KA test. One hour (Ph group) or thirty minutes after the last injection the convulsant KA was i.p. applied at a dose of 30 mg/kg. The intensity of seizures was scored each hour up to 200 min. During the first 20 min of observation, the KA injection induced mild seizure behavior consisting mainly of facial automatisms and head nodding. Furthermore, at about 40th minutes, the seizure intensity progressed to forelimb clonus and rearing with occasional loss of posture (score 3–4). That behavioral reaction was sustainable until 140 min and faded out gradually till 200 min of observation in the control group (Figure 2A). No significant difference in each time interval as well as total seizure intensity was detected between the Ph group and the veh group (Figure 2A,B). Significantly lower seizure scores were demonstrated in the *cis*-form of the **SB1-Ph** compound in the 80th (*p* < 0.05), 120th (*p* < 0.001), 140th (*p* < 0.001) and 160th min (*p* < 0.001), respectively, compared to veh group (Figure 2A). The *cis*-form of **SB4-Ph** compound alleviated SE at the 140th (*p* < 0.001) and 160th minute (*p* < 0.01), respectively, compared to the veh group.

### 2.4. Effects of Cis Isomers of SB1-Ph and SB4-Ph Derivates on the KA-Induced Oxidative Stress

The antioxidant capacity of the two *cis* isomers of SB1-Ph and SB4-Ph was assessed by measurement of the level of total glutathione (GSH) and lipid peroxidation in the hippocampus after the KA-induced SE in mice. A significant decrease in the total GSH was detected in the KA-veh group compared to the controls (*p* < 0.001) (Figure 3A). The *cis* isomer of SB4-Ph significantly elevated the level of endogenous antioxidant in the homogenate (*p* < 0.001 compared to the KA + veh group). The antioxidant activity of this derivate was comparable to the effect of phenytoin (*p* < 0.001 compared to the KA-veh group), while the *cis* isomer of SB1-Ph was ineffective (*p* > 0.05).

Furthermore, the KA + veh group showed elevated malondialdehyde (MDA) level in the hippocampus (*p* < 0.001 compared to C-veh group) suggesting an enhanced lipid peroxidation as a result of SE (Figure 3B). Neither the phenytoin nor the two *cis* isomers of the new Schiff bases succeeded to suppress the KA-induced lipid peroxidation in the hippocampus though the two phenytoin analogs partly mitigated this process (*p* < 0.01 compared to KA + veh group).

## 3. Discussion

Our findings revealed that the *trans/cis* conversion of 5,5′-diphenylhydantoin Schiff bases has protective activity against seizure spread in the MES test and mitigated the KA-induced SE. The antioxidant potency of *cis* **SB4-Ph** might be associated with its efficacy in reducing the severity of SE.

Two new 3-amino-5,5′-diphenylhydantoin Schiff Bases (**SB1-Ph** and **SB4-Ph**) were synthesized as described in detail in [10]. The compounds **SB1-Ph**, and **SB4-Ph** were synthesized by a condensation reaction in absolute methanol between 3-amino-5,5′-diphenylimidazolidine-2,4-dione (1) and the corresponding aromatic aldehyde: thiophene-2-carbaldehyde (2) or pyridine-2-carbaldehyde (3) in a 1:1 molar ratio in the presence of catalytic quantities of glacial acetic acid (Figure 1). Heterocycles thienyl, respectively pyridyl moiety in the **SB1-Ph** and **SB4-Ph** give the electron-donating properties of the molecules.

Recently [10] have been studying ground state DFT calculations as *trans*-isomers (**SB1-Ph** and **SB4-Ph**), supported by X-ray investigation, which has revealed a near planar shape around the -CH=N- bond. The *cis*-isomers are distinguished by their twisting shape and the creation of a weak noncovalent interaction. Azomethine aromatic compounds that can make weak noncovalent interactions with hydantoin rings in polar solvents play a critical function. Therefore, it was important to study and compare the anticonvulsant activity of both isomers *trans*- and *cis*- and to show that the stereoisomeric and conformation states of the molecules play an important role and possess different activity.

We found that compared to the *trans* isomers, the *cis* isomers of the two phenytoin Schiff Bases **SB1-Ph** and **SB4-Ph** exerted higher potency to suppress seizure spread in the MES test, which is consistent with earlier bioactivity investigations [11,12,13,14]. Furthermore, unlike phenytoin, the sub-chronic pretreatment with the two *cis* isomers of these Schiff Bases mitigated the severity of the KA-induced SE in mice. The detected potency of the *cis*-form of the **SB4-Ph** analog to elevate the total level of GSH and partly to reduce the lipid peroxidation in the hippocampus, suggest that the potency of this drug to minimize seizure severity during SE is closely related to its antioxidant activity in the hippocampus. However, the seizure-suppressing effect of **SB1-Ph** analog during SE was not accompanied by mitigation of oxidative stress, suggesting a difference from the **SB1-Ph** mechanism of its anticonvulsant effect. 

Azomethine aromatic compounds SB1-Ph and SB4-Ph have a donor thiophene/pyridine ring that can give favorable lipophylic interactions with the corresponding receptors. **SB1-Ph** contains the large S atom’s sterical hindrance in the molecule’s variable ring part, as opposed to **SB4-Ph** which contains an N atom. The enhanced biological activity of the *cis*-form in comparison to the *trans*-form, on the other hand, may also be due to the better conformational states and matches with the target receptors. We hypothesize that the underlying mechanism of the anticonvulsant activity of the two 5,5′-diphenylhydantoin Schiff bases, which are structurally similar to phenytoin, is different [11]. The two heterocyclic substituents, in particular, can be the molecule’s key pharmacophore.

## 4. Materials and Methods

### 4.1. Chemicals and Instrumentation

Each one of the chemicals and solvents was analytical or HPLC quality, bought from Fluka or Merck, and utilized unpurified. The 3-amino-phenytoin Schiff base derivatives: 5,5-diphenyl-3-((thiophen-2-ylmethylene)amino)-imidazolidine-2,4-dione (SB1-Ph), and 5,5-diphenyl-3-((pyridin-2-ylmethylene)amino)imidazolidine-2,4-dione (SB4-Ph) have been prepared by our recently described procedure [10]. The physicochemical and analytical data of the compounds were identical to those previously described. The *trans*/*cis* isomerization upon long wavelength UV light at 365 nm and *cis/trans* relaxation at room temperature is demonstrated in Figure 4.

### 4.2. Experimental Rodents

Male albino ICR mice (23–26 g), delivered by the vivarium of the Institute of Neurobiology-BAS, were left undisturbed for seven days before experimental procedures. The rodents were kept in transparent cages (10 in groups), with standard pellets and tap water ad libitum and in an artificial light–dark cycle regime (12:12; light on at 07:00 a.m.), T^o^ = 21 ± 1 °C and humidity: 40 ± 5%). The experiments were conducted in the morning (10:00–11:00 p.m.). All performed manipulations were consistent with the Declaration of Helsinki Guiding Principles on Care and Use of Animals (DHEW Publication, NHI 80–23) and with EC Directive 2010/63/EU for animal experiments. The experimental procedures were approved by the Bulgarian Food Safety Agency (License No: 354/2023). 

### 4.3. Experimental Design

A description of experimental groups and consequent procedural steps is described Figure 5. In short, the mice were allocated to two main experimental protocols. In Experiment#1, sixteen groups were used as follows: C group (control group, injected with a vehicle) (*n* = 16); Ph group (positive control, injected with phenytoin in 20 mg/kg) (*n* = 12); three groups, injected with *trans*-form of SB1-Ph in doses of 10, 20 and 40 mg/kg, respectively) (12 × 3); three groups, injected with *cis*-form of SB1-Ph in doses of 10, 20 and 40 mg/kg, respectively) (12 × 3); three groups, injected with *trans*-form of SB4-Ph in doses of 10, 20 and 40 mg/kg, respectively) (12 × 3); three groups, injected with *cis*-form of SB4-Ph in doses of 10, 20 and 40 mg/kg, respectively) (12 × 3). In Experiment#2, five groups were used as follows: C-veh group (control group, treated with a vehicle for 7 days) (*n* = 8); KA + Ph group (positive control, treated with phenytoin at a dose of 20 mg/kg for 7 days) (*n* = 8); KA + SB1-Ph groups (experimental group treated with *cis*-form of SB1-Ph at a dose of 40 mg/kg) (8); KA + SB4-Ph groups (experimental group treated with *cis*-form of SB4-Ph at a dose of 40 mg/kg) (8).

### 4.4. Drugs and Treatment

The 3-amino-phenytoin Schiff bases: (E)-5,5-diphenyl-3-((thiophen-2-ylmethylene)amino)- imidazolidine-2,4-dione (SB1-Ph) and (E)-5,5-diphenyl-3-((pyridin-2-ylmethylene)amino)imidazolidine-2,4-dione (SB4-Ph) have been prepared as described in our previous study [10]. Their *cis* isomers were freshly prepared before each experiment after UV irradiation exposure at λ = 365 nm to the probe *trans* isomers for 60 min. The compounds and the positive control phenytoin were dissolved in 1% DMSO before each test. Phenytoin was applied intraperitoneally (i.p.) an hour before the grip strength, rotarod, and MES at 30 mg/kg. The two Schiff bases were administered in three doses of 10, 20 and 40 mg/kg 0.5 h before the grip strength, rotarod and MES. The effective dose against tonic-clonic seizures of 40 mg/kg was applied in a sub-chronic regime of 7 days before KA (30 mg/kg., i.p.)-induced SE. The convulsant was administered 0.5 h after the last drug/vehicle injection.

### 4.5. Rotarod Test

The motor coordination was assessed as previously described [12]. The inability to keep position on a rotating rod (3.2 cm in diameter, at a speed of 10 rpm) for at least of one minute out of three trials was accepted as a criterion for neurotoxicity (>half of mice per group with lost balance). 

### 4.6. Muscle Strength

The grip strength device, attached to the dynamometer (BIOSEB, Chaville, France), was used to determine the muscle strength of each mouse. The animal was pulled backward by the tail after a tough grasp of the steel wire grid (8 cm × 8 cm) via forepaws. The average grasping force, expressed in N (newtons) ± S.E.M., of three trials was assessed for every animal.

### 4.7. Anticonvulsant Activity

#### 4.7.1. Maximal Electroshock (MES) Test

Corneal electroshock (50 mA, 60 Hz, 0.2 s) via electrodes (Constant Current Shock Generator) was applied to mice in the control group (C), phenytoin group (Ph), three SB1-Ph-treated with *trans* isomer groups as follows: injected with 10 mg/kg (SB1-Ph 10), (SB1-Ph 20) and (SB1-Ph 40), three SB1-Ph-treated with *cis* isomer groups as follows: injected with 10 mg/kg (SB1-Ph 10), (SB1-Ph 20) and (SB1-Ph 40), three SB4-Ph-treated with *trans* isomer groups as follows: injected with 10 mg/kg (SB4-Ph 10), (SB4-Ph 20) and (SB4-Ph 40) and three SB4-Ph-treated with *cis* isomer groups as follows: injected with 10 mg/kg (SB4-Ph 10), (SB4-Ph 20) and (SB4-Ph 40), respectively. Each group consisted of 6–7 mice. The controls exhibited tonic-clonic seizures with 100% of the hind limb tonic extensor component. The lack of an extensor component (forelimb tonic or only clonic seizures) was accepted as an anticonvulsant activity of the treatment.

#### 4.7.2. Kainate-Induced Status Epilepticus

The mice were assigned in groups of 8 mice. On the day of the 7th i.p. injection of the drug/vehicle, each animal was i.p. administered by 30 m/kg KA (FOT, Bulgaria). The convulsant was dissolved in saline (10 mL/kg of body weight). The observation of seizure onset and its intensity was scored according to the scale [15] as follows: stage 1 (facial clonus), stage 2 (nodding), stage 3 (forelimb clonus), stage 4 (forelimb clonus with rearing), and stage 5 (rearing and lost posture). The SE was characterized by continuous clonic seizures of stage 4 or 5. The seizures of the highest score detected during every 20 min up to 200 min were evaluated. Immediately after a 3 h period of observation, the mice were decapitated, brains were dissected rapidly on ice and the two hippocampi were isolated, quickly frozen in liquid nitrogen and stored at −20 °C until ELISA analysis. 

### 4.8. Measurement of Glutathione (GSH) and Malondialdehyde (MDA) in the Hippocampus

The isolated hippocampi were kept on ice, weighed, and preserved at 20 °C until homogenization in cold PBS buffer (pH 7.4) containing 1 mM EGTA, 50 mM NaF, 1 mM 270 EDTA and 1 mM PMSF. After centrifugation of the tissue homogenate at 5000× *g*, 4 °C for 271 10 min, GSH and MDA were measured in duplicates using an ELISA kit (Elabscience cat. 272 No E-EL-0060 and E-EL-0026) according to the manufacturer’s instructions. The results (273) were expressed in µg/mg protein (GSH) and ng/mg protein (MDA).

### 4.9. Data Analysis

The data are expressed as means ± SEM. The results from the MES test and rotarod test 315 were analyzed by Fisher’s exact test. Data from the grip strength test, KA test and bio-316 chemistry were assessed by the one-way analysis of variance (ANOVA) followed by the 317 post hoc Bonferroni’s test. *p* < 0.05 was considered statistically significant.

## 5. Conclusions

In conclusion, the *cis* isomers of 3-amino-5,5′-diphenylhydantoin Schiff Bases (SB1-Ph and SB4-Ph) exhibited higher potency than their *trans*-forms to suppress seizure spread and tonic seizures in mice. The anticonvulsant activity of the *cis* isomer SB4-Ph against the neurotoxin KA might be associated with the antioxidant potency in the hippocampus during SE.

## Data Availability

Data are contained within the article.

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
