# Peer review of "Comparative Analysis of Anticonvulsant Activity of Trans and Cis 5,5′-Diphenylhydantoin Schiff Bases"

_ijms, 2023, doi:10.3390/ijms242216071_

Round 1

Reviewer 1 Report

Comments and Suggestions for Authors

This study is a very basic evaluation of the effect of several newly synthesized Schiff bases for anticonvulsant activity in combination with phenytoin.

I have few questions and suggestions for the authors before it can be accepted for publication.

1.       In Table 1, the last column represents Rotarod test (N/F). Please define how this was calculated and what are these values. This is not well explained in the text.

2.       What statistical test was performed to measure the significance of Figure 1? Please include that information.

3.       What is total seizure intensity and how it was quantified?

4.       Please define/mention the full name of GSH and MDA before using these terms as abbreviations.

Author Response

Thank you for the careful evaluation of our manuscript. We have revised the manuscript, taking into account the suggested modifications. All changes in the MS are highlighted in red.

Point #1 This study is a fundamental evaluation of the effect of several newly synthesized Schiff bases for anticonvulsant activity in combination with phenytoin.I have few questions and suggestions for the authors before it can be accepted for publication. In Table 1, the last column represents Rotarod test (N/F). Please define how this was calculated and what are these values. This is not well explained in the text.

Response: Response: Following the Referee's advice, a description of how the results from the rotarod test were calculated is presented in the text of Table 1.

Point # 2 What statistical test was performed to measure the significance of Figure 1? Please include that information.

Response: We’re thankful for this relevant remark—additional subsection in 4. Materials and Methods section (4.9. Data Analysis) was inserted. All statistical methods applied for each test was described

Point # 3 What is total seizure intensity and how it was quantified?

Response: The seizures of the highest score detected every 20 min up to 200 min were evaluated for each mouse in the group according to the scale outlined by Racine. Therefore, data for seizure intensity were calculated as mean±SEM for each observation time interval.

Point # 4 Please define/mention the full name of GSH and MDA before using these terms as abbreviations.

Response: Following the Reviewer's advice, when mentioned first, the full names of GSH and MDA were given.

Reviewer 2 Report

Comments and Suggestions for Authors

The manuscript “Comparative analysis of anticonvulsant activity of trans and cis 5,5’-diphenylhydantoin Schiff bases” is devoted to the investigations of anticonvulsant activity of two forms of hydantoin based hydrazones. The research is comprehensive and is of high practical significance. The authors have found that cis forms are more active than trans ones, which is very interesting for the development of new anticonvulsant drugs. Despite interesting concept, findings, novelty and practical importance, the manuscript has several serious experimental flows in chemistry.

In my opinion, this manuscript suits to the scope of IJMS. I recommend that after major revision, it can be accepted.

Comments for the authors:

1. Line 53: VV-hemorphin-5 – an illustration with chemical formula is required.

2. For new compounds, physicochemical characterization data must be added both to the manuscript and supporting information (NMR plots).

3. A chemical procedure for the preparation of cis forms along with their physicochemical characterization data must be added to the manuscript.

Author Response

Thank you for the careful evaluation of our manuscript. We have revised the manuscript taking into account the suggested modifications. All changes in the MS are highlighted in red.

Point #1 1. Line 53: VV-hemorphin-5 – an illustration with chemical formula is required.

Response: We‘re thankful for this relevant note. The sentence was modified because it was unclear. Recently, the four 5,5’-diphenylhydantoin Schiff bases containing aromatic species (SB1–SB4) were synthesized by our team (Todorov, P.; Georgieva, S.; Peneva, P.; Rusew, R.; Shivachevc, B.; Georgiev, A. Experimental and theoretical study of bidirectional photoswitching behavior of 5,5′-diphenylhydantoin Schiff bases: synthesis, crystal structure and kinetic approaches. New J. Chem. 2020, 44, 15081-15099). In addition, their structure–property characterization was carried out by X-ray, optical, and electrochemical methods (Todorov et al., 2020). Further, the four Schiff bases of phenytoin analogs were studied for potential anticonvulsant activity (Tchekalarova et al., 2023). In the present study, a comparison between the trans- and the cis isomers of the most active two compounds was conducted.

Point # 2. For new compounds, physicochemical characterization data must be added both to the manuscript and supporting information (NMR plots).

Response: As we mentioned in Point#1, the newly synthesized 5,5’-diphenylhydantoin Schiff bases were characterized already, including their physicochemical properties were reported in our previous manuscript (https://pubs.rsc.org/en/content/articlelanding/2020/NJ/D0NJ03301D; https://www.rsc.org/suppdata/d0/nj/d0nj03301d/d0nj03301d1.pdf

(„The 3-amino-phenytoin Schiff base derivatives: 5,5-diphenyl-3-((thiophen-2-ylmethylene)amino)-imidazolidine-2,4-dione (SB1-Ph), and 5,5-diphenyl-3-((pyridin-2-ylmethylene)amino)imidazolidine-2,4-dione (SB4-Ph) have been prepared by our recently described procedure by [12]. The physicochemical and analytical data of the compounds were identical to those previously described [12].”)

Point # 3 A chemical procedure for the preparation of cis forms along with their physicochemical characterization data must be added to the manuscript.

Response: The procedure for the preparation of cis forms and data related to their physicochemical characteristic are presented in 4. Materials and Methods and Figure 4.

(“…The 3-amino-phenytoin Schiff base derivatives: 5,5-diphenyl-3-((thiophen-2-ylmethylene)amino)-imidazolidine-2,4-dione (SB1-Ph), and 5,5-diphenyl-3-((pyridin-2-ylmethylene)amino)imidazolidine-2,4-dione (SB4-Ph) have been prepared by our recently described procedure by [12]. The physicochemical and analytical data of the compounds were identical to those previously described [12]. The trans/cis isomerization upon long wavelength UV light at 365 nm and cis/trans relaxation at room temperature is demonstrated in Fig. 4.”).

Reviewer 3 Report

Comments and Suggestions for Authors

The article  ‘’Comparative analysis of anticonvulsant activity of trans and cis 5,5’-diphenylhydantoin Schiff bases ’’ focuses on the synthesis, characterization, and evaluation of four cis and trans 5,5’-diphenylhydantoin Schiff bases for anticonvulsant activity in combination with phenytoin.

Sentence structure of few sentences is not up to the mark as there are several punctuation and grammar related mistakes.

Spectroscopic characterization of synthesized compounds has not been provided. There is no spectroscopic data related discussion provided in the manuscript. It is mandatory to provide related spectra of spectroscopic techniques (MS, NMR spectroscopy) and elemental analysis to validate the synthesis of  5,5’-diphenylhydantoin Schiff bases. Moreover, the spectroscopic data must be provided with details in the manuscript.

Moreover, structure activity relationship should be discussed to emphasize the role of substituents towards anti-convulsant activity.

Generally, all the references are appropriate, however, some irregularities have been found. As most of the references are given with doi numbers. However, doi number is missing in few references. Keep same format throughout the manuscript.

Other Minor Remarks:

Ø  Sentence between line 15 & 16 needs to be re-written.

Ø  Line 18 and 68, add ‘’the’’ before word ‘’doses’’.

Ø  Sentence format of line 20 and 21 needs to be revised.

Ø  Line 45, replace word ‘’increase’’ with ‘’increased’’.

Ø  Line 63, add comma after kg.

Ø  In table values, make sure the use of comma or full stop in between the grip strength values.

Ø  Line 106, add comma after word ‘’observation’’.

Ø  Line 108, omit article ‘’a’’.

Ø  Sentence within the line 154 & 155, needs  to be re-written for clear understanding.

Ø  Line 185, omit word ‘’by’’.

Ø  Figure 4, conditions on arrow should not exceed the arrow head.

Ø  Line 197, write degree symbol as ‘’oC’’.

Ø  Line 198, correct the spelling of ‘’with’’.

Ø  Line 226, replace ‘’refrant’’ with ‘’refrence’’

Ø  In Section, Measurement of GSH and MDA in the Hippocampus, correctly add degree sign.

Ø  Line 277, correct spelling of ‘’isomers’’.

Comments on the Quality of English Language

A thorough critical proof-reading of manuscript is suggested to avoid spelling and grammar related mistakes.

Author Response

Thank you for the careful evaluation of our manuscript. We have revised the manuscript taking into account the suggested modifications. All changes in the MS are highlighted in red.

Point #1   Sentence structure of few sentences is not up to the mark as there are several punctuation and grammar related mistakes.

Response: We’re thankful for this relevant remark of the Reviewer. We have read the whole text carefully and tried to correct all technical errors and inappropriate words and sentences.

Point #2   Spectroscopic characterization of synthesized compounds has not been provided. There is no spectroscopic data related discussion provided in the manuscript. It is mandatory to provide related spectra of spectroscopic techniques (MS, NMR spectroscopy) and elemental analysis to validate the synthesis of  5,5’-diphenylhydantoin Schiff bases. Moreover, the spectroscopic data must be provided with details in the manuscript.

Response: We‘re thankful for this relevant note. The sentence was modified because was unclear. Recently, the four 5,5’-diphenylhydantoin Schiff bases, containing aromatic species (SB1–SB4) were synthesized by our team (Todorov, P.; Georgieva, S.; Peneva, P.; Rusew, R.; Shivachevc, B.; Georgiev, A. Experimental and theoretical study of bidirectional photoswitching behavior of 5,5′-diphenylhydantoin Schiff bases: synthesis, crystal structure and kinetic approaches. New J. Chem. 2020, 44, 15081-15099). In addition, their structure–property characterization was carried out by X-ray, optical and electrochemical methods (Todorov et al., 2020). Further, the four Schiff bases of phenytoin analogs were studied for potential anticonvulsant activity (Tchekalarova et al., 2023). In the present study a comparison between the trans- and the cis isomers of the most active two compounds was conducted.

Please, refer to: https://pubs.rsc.org/en/content/articlelanding/2020/NJ/D0NJ03301D

Point #3    Moreover, structure activity relationship should be discussed to emphasize the role of substituents towards anti-convulsant activity.

Response: We are thankful for this relevant remark. In the last paragraph of the Discussion section, the structure-activity relationship was discussed.

Point # 4 Generally, all the references are appropriate, however, some irregularities have been found. As most of the references are given with doi numbers. However, doi number is missing in few references. Keep same format throughout the manuscript.

Response: All missing information related to doi of the cited references was added in the new version.

Point # 5 Other Minor Remarks: Ø  Sentence between line 15 & 16 needs to be re-written.

Response: Corrected.

Point # 6 Ø  Line 18 and 68, add ‘’the’’ before word ‘’doses’’.

Response: Corrected.

Point # 7 Ø  Sentence format of line 20 and 21 needs to be revised.

Response: Corrected.

Point # 8 Ø  Line 45, replace word ‘’increase’’ with ‘’increased’’.

Response: Corrected.

Point # 8 Ø  Line 63, add comma after kg.

Response: Corrected.

Point # 9 Ø  In table values, make sure the use of comma or full stop in between the grip strength values.

Response: Corrected.

Point # 10 Ø  Line 106, add comma after word ‘’observation’’.

Response: Corrected.

Point # 11 Ø  Line 108, omit article ‘’a’’.

Response: Corrected.

Point # 12 Ø  Sentence within the line 154 & 155, needs  to be re-written for clear understanding.

Response: Corrected.

Point # 13 Ø  Line 185, omit word ‘’by’’.

Response: Corrected.

Point # 14 Ø  Figure 4, conditions on arrow should not exceed the arrow head.

Response: Corrected.

Point # 15  Ø  Line 197, write degree symbol as ‘’oC’’.

Response: Corrected.

Point # 16 Ø  Line 198, correct the spelling of ‘’with’’.

Response: Corrected.

Point # 17 Ø  Line 226, replace ‘’refrant’’ with ‘’refrence’’

Response: Corrected.

Point # 18 Ø  In Section, Measurement of GSH and MDA in the Hippocampus, correctly add degree sign.

Response: Corrected.

Point # 19 Ø  Line 277, correct spelling of ‘’isomers’’.

Response: Corrected.

Point # 20 Comments on the Quality of English Language A thorough critical proof-reading of manuscript is suggested to avoid spelling and grammar related mistakes.

Response: We are thankful to the Reviewer for all relevant notes. We have read the whole text carefully and tried to correct all technical errors and inappropriate words and sentences.

Reviewer 4 Report

Comments and Suggestions for Authors

After reading the manuscript my major concerns are as follows:

  1. The manuscript needs an intensive correction of the English syntax and grammar.
  2. Please, replace the term “antiepileptic drugs” with “antiseizure medications” – the term recommended by International League Against Epilepsy.
  3. Abstract – line 21: Please, delete the second duplication in this sentence … “in the hippocampus.”
  4. Table 1. Decimal system in English is based on a full stop. Please, replace commas with full stops.
  5. Since PHT has its time to peak anticonvulsant effect established at 120 min (see: Loscher et al., 1991), why the tested Schiff bases of PHT (SB1-PH and SB4-Ph) were administered at 15 min. before the MES-induced seizures?
    --- Löscher W, Fassbender CP, Nolting B. The role of technical, biological and pharmacological factors in the laboratory evaluation of anticonvulsant drugs. II. Maximal electroshock seizure models. Epilepsy Res. 1991 Mar;8(2):79-94. doi: 10.1016/0920-1211(91)90075-q.
  6. Please, insert information on the test applied to statistically compare the results from the grip-strength test?
  7. Are the authors sure that the units from the grip-strength test are in Newton? Perhaps, gram-force?
  8. Line 98: please delete the duplication: “were pretreated”
  9. Figure 3. Please insert explanation of symbols used in the Figure 3b – two asterisks and one zero
  10. Please correct the sentence on line 204. “A description … is described”???
  11. Please provide information about the percent of cis-forms present in the mixture of Schiff bases Ph after 1 h of exposure to the UV light. Was there a 100%? Provide info on the percent of trans-forms present in the mixture. Since the visible light and room temperature reversed cis-forms into trans-forms, what kind of equilibrium between both forms was observed in each Schiff bases of Ph? Do really all trans-isoforms of Schiff bases Ph convert to their cis- counterparts?
  12. Section 4.6 – Muscle strength – the name of the company producing the device for testing skeletal strength in rodents is “BIOSEB”. Please correct it.
  13. Line 259 - Please correct the unit in the dose of KA.
  14. Please correct the third affiliation – it should be: Medical.
Comments on the Quality of English Language

See my previous comments

Author Response

Thank you for the careful evaluation of our manuscript. We have revised the manuscript taking into account the suggested modifications. All changes in the MS are highlighted in red.

Point #1 The manuscript needs an intensive correction of the English syntax and grammar.

Response: We are thankful to the Reviewer for all relevant notes. We have read the whole text carefully and tried to correct all technical errors and inappropriate words and sentences.

Point # 2 Please, replace the term “antiepileptic drugs” with “antiseizure medications” – the term recommended by International League Against Epilepsy.

Response: We’re thankful for this note. The AEDs were replaced in a whole new version with the newly accepted nomenclature ASMs. So far, there is no cure for epilepsy, and existing medications primarily focus on symptom management, particularly in controlling seizures.

Point #  3 Abstract – line 21: Please, delete the second duplication in this sentence … “in the hippocampus.”

Response: Corrected

Point # 4 Table 1. Decimal system in English is based on a full stop. Please, replace commas with full stops.

Response: Corrected.

Point #5 Since PHT has its time to peak anticonvulsant effect established at 120 min (see: Loscher et al., 1991), why the tested Schiff bases of PHT (SB1-PH and SB4-Ph) were administered at 15 min. before the MES-induced seizures?
--- Löscher W, Fassbender CP, Nolting B. The role of technical, biological and pharmacological factors in the laboratory evaluation of anticonvulsant drugs. II. Maximal electroshock seizure models. Epilepsy Res. 1991 Mar;8(2):79-94. doi: 10.1016/0920-1211(91)90075-q.

Response: As we mentioned in the text, “…the four 3-amino-phenytoin Schiff base derivatives were explored alone and in combination with phenytoin against MES-induced seizure spread in mice [1 3]. The docking analysis suggests that although structurally similar to phenytoin the underlying mechanism of the anticonvulsant activity of the two 5,5'-diphenylhydantoin Schiff bases might be different [13]. The preliminary screening with these new Schiff bases revealed that they have peak anticonvulsant activity 0.5 h (30 minutes) before i.p. injection.

Point #6 Please, insert information on the test applied to statistically compare the results from the grip-strength test?

Response: We’re thankful for this relevant remark. Additional subsection in 4. Materials and Methods section (4.9. Data Analysis) was inserted. All statistical methods applied for each test was described, including the results for the grip-strength test.

Point # 7 Are the authors sure that the units from the grip-strength test are in Newton? Perhaps, gram-force?

Response: Grip strength is typically measured in pounds, kilograms, or Newtons. Our device (https://www.bioseb.com/en/activity-motor-control-coordination/48-grip-strength-test.html) has Sensor Capacity 0 – 2, 5 kg (25N).

Point # 8 Line 98: please delete the duplication: “were pretreated”

Response: Corrected.

Point # 9 Figure 3. Please insert explanation of symbols used in the Figure 3b – two asterisks and one zero

Response: The missing explanation of symbols in Fig. 3B was inserted in the text to the Figure.

Point # 10 Please correct the sentence on line 204. “A description … is described”???

Response: It was corrected. Thank you.

Point # 11 Please provide information about the percent of cis-forms present in the mixture of Schiff bases Ph after one h of exposure to the UV light. Was there a 100%? Provide info on the percent of trans-forms present in the mixture. Since the visible light and room temperature reversed cis-forms into trans-forms, what kind of equilibrium between both forms was observed in each Schiff base of Ph? Do really all trans-isoforms of Schiff bases Ph convert to their cis- counterparts?

Response: The 3-amino-phenytoin Schiff base derivatives: 5,5-diphenyl-3-((thiophen-2-ylmethylene)amino)-imidazolidine-2,4-dione (SB1-Ph), and 5,5-diphenyl-3-((pyridin-2-ylmethylene)amino)imidazolidine-2,4-dione (SB4-Ph) have been prepared by our recently described procedure [12]. The physicochemical and analytical data of the compounds were identical to those previously described [12]. The trans/cis isomerization upon long wavelength UV light at 365 nm and cis/trans relaxation at room temperature is demonstrated in Fig. 4.

Point # 12 Section 4.6 – Muscle strength – the name of the company producing the device for testing skeletal strength in rodents is “BIOSEB”. Please correct it.

Response: It was corrected.

Point #13 Line 259 - Please correct the unit in the dose of KA.

Response: Corected.

Point #14 Please correct the third affiliation – it should be: Medical.

Response: Corrected.

Point #15 Comments on the Quality of English Language See my previous comments

Response: We have read the whole text carefully and tried to correct all technical errors and inappropriate words and sentences.

Round 2

Reviewer 2 Report

Comments and Suggestions for Authors

The authors provided neccesary changes. The manuscript is ready for acceptance. 

Author Response

We are thankful for the valuable comments!

Reviewer 3 Report

Comments and Suggestions for Authors

The abstract fails to give clear information.

In line 46, the term "they have" has been used for compounds, correct it.

The lines 15, 16 and 169 are not conveying any clear information.

The detailed structural activity relationship should be provided separately. 

In figure 4, conditions are exceeding the arrow head. 

In line 197 and 307, the degree sign should be superscript.

The doi number in reference 5, 6 and 7 should be given according to the format of whole manuscript.

Comments on the Quality of English Language

English language should be carefully checked.

Author Response

Pont # 1: In line 46, the term "they have" has been used for compounds, correct it.

Response: Corrected.

Pont # 2:The lines 15, 16 and 169 are not conveying any clear information.

Response: We agree with the Reviewer remark. The text was edited.

Pont # 3: The detailed structural activity relationship should be provided separately.

Response: Corrected.

Pont # 4: In figure 4, conditions are exceeding the arrow head. 

Response: The figure was edited in the new version.

Pont # 5: In line 197 and 307, the degree sign should be superscript.

Response: Corrected.

Pont # 6: The doi number in reference 5, 6 and 7 should be given according to the format of whole manuscript.

Response: Only references with doi number were given

Reviewer 4 Report

Comments and Suggestions for Authors

I have still two minor concerns:

1. Page 2 - please, insert into the text of the Grip-strength results section: F-statistics value with degrees of freedom and p values when describing the results from the grip-strength test.

example: [one-way Anova: F(2,45) = 3.456; p = 0.045]

2. Page 9, paragraph 4.6 Muscle strength. Please change the name of the company from BIOCEB to BIOSEB. It should be BIOSEB !

Comments on the Quality of English Language

No comments

Author Response

Comments and Suggestions for Authors

I have still two minor concerns:

Point #1:  Page 2 - please, insert into the text of the Grip-strength results section: F-statistics value with degrees of freedom and p values when describing the results from the grip-strength test. example: [one-way Anova: F(2,45) = 3.456; p = 0.045]

Response: This statistical data were inserted in the text to the new version.

Point# 2: Page 9, paragraph 4.6 Muscle strength. Please change the name of the company from BIOCEB to BIOSEB. It should be BIOSEB !

Response: Corrected.